# Sexually divergent expression of active and passive conditioned fear responses in rats

Tina M Gruene[1], Katelyn Flick[1], Alexis Stefano[1], Stephen D Shea[2], Rebecca M Shansky[1]*

[1]Department of Psychology, Northeastern University, Boston, United States; [2]Cold Spring Harbor Laboratories, Cold Spring Harbor, United States

**Abstract** Traditional rodent models of Pavlovian fear conditioning assess the strength of learning by quantifying freezing responses. However, sole reliance on this measure includes the de facto assumption that any locomotor activity reflects an absence of fear. Consequently, alternative expressions of associative learning are rarely considered. Here we identify a novel, active fear response ('darting') that occurs primarily in female rats. In females, darting exhibits the characteristics of a learned fear behavior, appearing during the CS period as conditioning proceeds and disappearing from the CS period during extinction. This finding motivates a reinterpretation of rodent fear conditioning studies, particularly in females, and it suggests that conditioned fear behavior is more diverse than previously appreciated. Moreover, rats that darted during initial fear conditioning exhibited lower freezing during the second day of extinction testing, suggesting that females employ distinct and adaptive fear response strategies that improve long-term outcomes.

*For correspondence: r.shansky@
neu.edu

Competing interests: The
authors declare that no
competing interests exist.

Reviewing editor: Peggy
Mason, University of Chicago,
United States

## Introduction

In the laboratory, auditory or "cued" fear conditioning and extinction in rodents are the predominant tools for studying the neural mechanisms of learning and memory for aversive stimuli (*Blanchard and Blanchard, 1969*; *LeDoux, 2000*; *Maren, 2001*). In these assays, the strength of a tone-shock association is traditionally measured by the fraction of time during the conditioned stimulus (CS) that subjects exhibit freezing, defined as the cessation of all movement not required for respiration (*Fanselow, 1980*). Accordingly, low freezing is generally interpreted as reflecting a weak association and thus poor learning. Likewise, low freezing after extinction is taken to indicate successful suppression of the conditioned response, a new memory (*Quirk and Mueller, 2008*). However, by their construction, these traditional assays are insensitive to alternative expressions of fear, such as escape.

Most studies of fear conditioning and extinction in rodents use exclusively male subjects (*Lebron-Milad and Milad, 2012*). The few studies that directly compare conditioned freezing responses in males and females produced mixed results (*Shansky, 2015*) but most frequently reported lower freezing in females (*Gupta et al., 2001*; *Maren et al., 1994*; *Pryce et al., 1999*). Whether this effect reflects genuine learning deficits in females, or is related to sex differences in fear response strategies is unknown. For example, females reliably exhibit heightened ambulation in a wide variety of common behavioral tests (*Archer, 1975*; *Fernandes et al., 1999*; *File, 2001*; *Seney et al., 2012*), which may influence their selection of responses to threatening stimuli.

To identify possible alternative fear response strategies, we evaluated locomotor activity in gonadally intact adult male (n=56) and female (n=58) Sprague Dawley rats as they were trained and tested in auditory fear conditioning (5 habituation CS followed by 7 CS-US pairs), extinction (20 CS),

**eLife digest** Animals can respond to fear in a variety of ways, such as by standing still (freezing), or rapidly escaping from an apparent threat. Freezing is the most common measure of fear that has been used in research studies. However, since the vast majority of these experiments have used male animals, it is not clear if freezing is a sufficient measure of fear in females.

To address this question, Gruene et al. analyzed different types of fear responses in large groups of male and female rats. The experiments used a technique called cued fear conditioning, which pairs a sound with a mild electrical shock to a foot. When rats learn that the sound predicts the shock, the sound alone causes them to produce a fear response. However, if the sound is then played repeatedly without a footshock, the rats learn to become less fearful of the sound in another learning process called "extinction".

The experiments found that females were four times more likely than males to display fear in the form of rapid movements (referred to as "darting"). Animals that displayed darting were also less likely to freeze in response to the sound cue, which suggests that darting may represent an alternative fear strategy that is more common in females. During a subsequent extinction test, females that darted also displayed quicker reductions in both types of fear responses, which suggests that darting might be an active coping response that promotes long term reductions in fear.

Gruene et al.'s findings suggest that there are differences in the ways that males and females respond in fear of a threatening stimulus, and highlight the importance of analyzing a variety of fear responses in experiments. The next steps will be to understand the biological basis of the darting response in female rats.

and extinction retention (3 CS) tests across 3 days(*Gruene et al., 2015*) (*Figure 1a*). In many animals, we qualitatively observed a rapid 'darting' behavior during fear conditioning tone presentation–a rapid, forward movement across the chamber that resembled an escape-like response (illustrated in *Figure 1b*, and *Video 1*). We quantified these responses by identifying and counting them as discrete events in traces of each animal's velocity for all sessions using Noldus Ethovision software and custom Matlab code (see *source code 1*, Materials and methods, and *Figure 1c*). We calculated darting rate (darts/min) during four non-overlapping trial epochs: 1) 60 s pre-CS period, 2) 30 s CS presentation, 3) 5 s "shock response" period, and 4) 30 s post-shock period (*Figure 2a*). This approach allowed us first to determine if darting reflected an alternate conditioned response, and second, whether the expression of conditioned darting predicted distinct behavioral patterns across fear conditioning and extinction.

## Results and Discussion

Prior to the initiation of shocks, darts were not temporally structured with respect to the CS. However, we found that females, but not males, exhibited increased dart frequency in response to CS onset during late trials (*Figure 1c–e*), suggesting that darting is a learned response. *Figure 1d,e* represent dart frequency amongst entire female and male cohorts, respectively.

We next compared darting in males and females across all test sessions. Females exhibited higher CS dart rates than males on all 3 days (*Figure 2c,i,j*; conditioning: ns p=0.07, Mann Whitney test. p<0.001 2-way ANOVA main effect of sex, $F_{1,112}$=12.1; sex x trial interaction $F_{11,1232}$=2.12, p=0.02 Extinction: p=0.01, Mann Whitney test. p<0.05, 2-way ANOVA main effect of sex, $F_{1,112}$=4.05. Extinction test: p=0.008, Mann Whitney test (Pre-CS); p<0.001, 2-way ANOVA main effect of sex, $F_{1,112}$=14.58 (CS)). Notably, CS dart rate in females increased as CS-US presentations progressed (*Figure 2c*) and decreased during extinction (*Figure 2i*), again suggesting that darting may reflect an alternate expression of associative learning. During fear conditioning, although both males and females reliably darted during the shock response period (*Figure 2d*; p<0.01 2-way ANOVA main effect of sex, $F_{1,112}$=8.5), shock-evoked darts in females reached higher velocities than darts in males (*Figure 2e*; p<0.0001 2-way ANOVA main effect of sex, $F_{1,112}$=20.35). Additionally, females were more likely to dart during the 30s post-shock period than males (*Figure 2f*; p<0.0001 2-way ANOVA

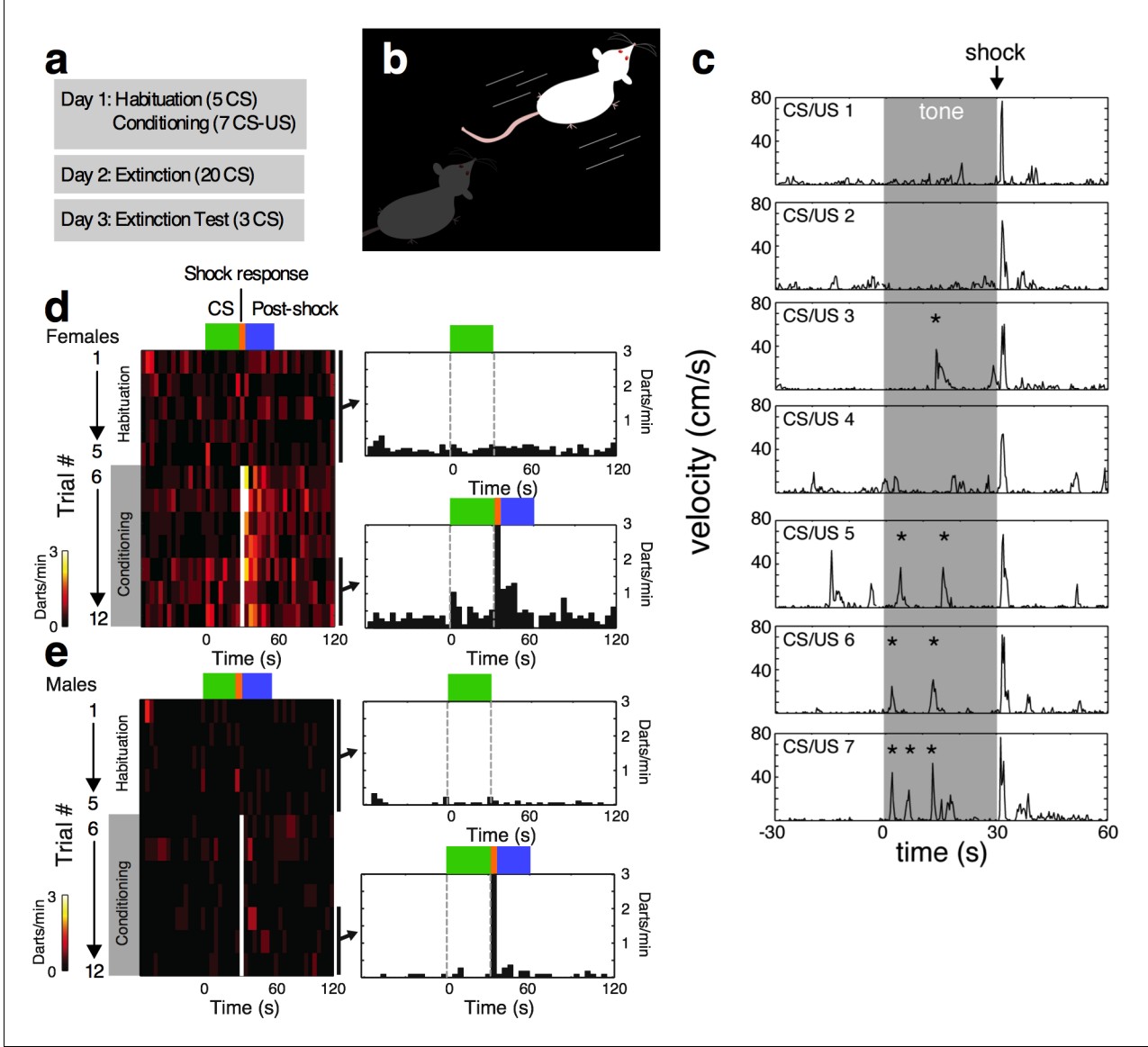

**Figure 1.** Darting is an active learned response to the CS that occurs primarily in female rats. (**a**) Experimental timeline. (**b**) Darts were characterized by a brief, high velocity movement across the test chamber. (**c**) Velocity traces from a representative animal, demonstrating increase in conditioned darting events across fear conditioning trials. Asterisks denote events that reached criterion for darting during the CS. Time 0 denotes CS onset. (**d**) Temporal organization of darting in all female rats. On the left is a two dimensional histogram of dart timing relative to the CS averaged over all females for 5 habituation trials and 7 conditioning trials on day 1. Trial time is on the x-axis and colored bars denote the trial epochs we defined as CS (green), shock response (orange), and post shock response (blue). Each row represents a CS trial (habituation 1–5, and conditioning 6–12), and depicts average dart rate by the color in each 4-second bin according to the color bar. On the right are histograms of the temporal organization of darts averaged over the five habituation trials (top) and the last three conditioning trials (bottom). Darts were detected and counted as described in Materials and methods. (**e**) Temporal organization of darting in all male rats. Panels are organized as in (**d**). During habituation trials, darts occurred at low rates throughout the trial in both sexes. In contrast, after conditioning only females exhibited increased darting triggered by tone onset ('CS') and sustained darting after shock delivery ('Post-shock'). Both sexes darted in response to the shock itself (Shock response). In both sexes, the first bin after the shock exceeds the limits of the y-axis.

main effect of sex, $F_{1,112}=23.27$). To determine whether an animal's shock response velocity was related to its overall propensity to dart, we correlated the mean velocity reached across all 7 US presentations with total detected darts during fear conditioning. These measures were significantly correlated in females (*Figure 2g*) but not males (*Figure 2h*), suggesting that in females only, an animal's immediate reaction to an aversive stimulus may influence its future response strategies.

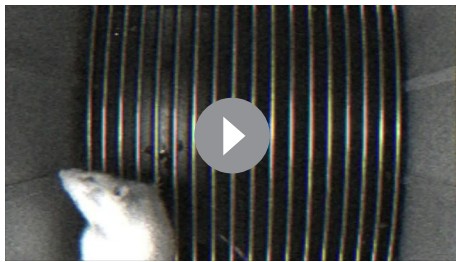

**Video 1.** Example of conditioned darting. A "Darter" during CS 12, corresponding to the last trace in *Figure 1C*. CS onset begins at 0:02 and continues through the entirety of the 18 sec video. The word "DART" appears on screen in red text during each of three observable darts.

We did not observe darting in all females, however, and so to identify possible behavioral markers and outcomes of darting, we divided animals into 'Darter' and 'Non-darter' sub-groups. An animal qualified as a "Darter" if it exhibited at least one dart during fear condition-ing tones (CS) 8–12. CS 8 is the 3rd CS-US pair-ing, and the point at which we usually observe a robust increase in freezing in males. Therefore, only darts that occurred during this same time period were considered to reflect conditioned darting. Over 40% of females qualified as Dar-ters (*Figure 3a*), whereas approximately 10% of males qualified (*Figure 3f*; chi-square = 13.8, p=0.0002). There was no effect of the estrous cycle on darting (*Figure 3* - supplement). Com-pared to Non-darters, female Darters exhibited greater shock response velocities (*Figure 3b*; p=0.001 2-way ANOVA main effect of group $F_{1,56}=11.49$), as well as higher dart rates in the post-shock period (*Figure 3c*; p<0.0001 2-way ANOVA main effect of group, $F_{1,56}=25.42$.), suggesting that female Darters have a more robust and protracted response to the shock. Importantly, female Darters did not exhibit higher dart rates during pre-CS periods or during CS-only habituation trials (*Figure 3d*; p=0.65, Mann Whitney test), suggesting that Darters are not simply more active overall, and were not pre-disposed to dart in response to the CS. During the CS, Darters exhibited increased darting as CS-US pairs progressed (p<0.0001 2-way ANOVA group x trial interaction, $F_{11,616}=8.8$; main effect of group $F_{1,56}=26.35$, p<0.0001). During the Extinction Pre-CS period, Darters did not dart more than Non-darters (p=0.38 Mann Whitney), but Darters exhibited increased darting during the first Extinction CS, suggesting that darting is a conserved conditioned response (2-way ANOVA interaction, $F_{19,1064}=1.584$, p=0.05; *p<0.05 Sidak's post-hoc test).

We next asked whether CS darting during fear conditioning related to CS freezing behavior (*Figure 3e*). In females, Darters and Non-darters did not differ in pre-CS or CS-only (habituation) freezing during fear conditioning. However, as CS-US pairings progressed, Darters froze less than Non-darters, suggesting that increased darting may prevent or compete with freezing responses p=0.02 2-way ANOVA group x trial interaction $F_{11,616}=2.16$. main effect of darting $F_{1,56}=4.18$, p<0.05). Darters and Non-darters did not significantly differ in freezing during Extinction. However, Darters also froze less during the extinction test (day 3; p<0.02, 2-way ANOVA main effect of group $F_{1,56}=5.76$) despite not exhibiting increased darting at that time, suggesting that darting during fear conditioning does not simply compete with an animal's freezing response, but may also reflect an adaptive response that predicts positive outcomes after extinction learning.

In the small subpopulation of male Darters, CS dart rate (*Figure 3i*; Conditioning p<0.0001, 2-way ANOVA group x trial interaction, $F_{11,594}=3.76$. Extinction: p<0.0001, 2-way ANOVA group x trial interaction, $F_{19,1026}=3.17$) and freezing (*Figure 3j*; Conditioning: p<0.01 2-way ANOVA group x trial interaction $F_{11,583}=2.68$. Extinction and extinction test: No significant interaction or effects) patterns during fear conditioning shared some characteristics with those in females.

However, there are several notable distinctions between male and female Darters. First, CS dart rate in darting males was characterized by a steady low rate of darting across trials (*Figure 3i*), instead of the increase across trials observed in females (*Figure 3d*), suggesting that darting in males may not reflect a learned fear response, but general hyperactivity that results in less freezing. Second, unlike our observations in females, male Darters did not exhibit heightened shock response velocities (*Figure 3g*) or robust post-shock dart rates (*Figure 3h*; p=0.01 2-way ANOVA group x trial interaction, $F_{6,324}=2.8$, no main effects) compared to Non-darters. Third, male Darters did not exhibit lower freezing during extinction testing, suggesting that the potential long-term behavioral implications of darting during fear conditioning are stronger in females than in males. Together with the large observed sex difference in darting prevalence (*Figure 2a,f*), these discrepancies suggest that there may be qualitative differences in the potential causes and effects of darting in males

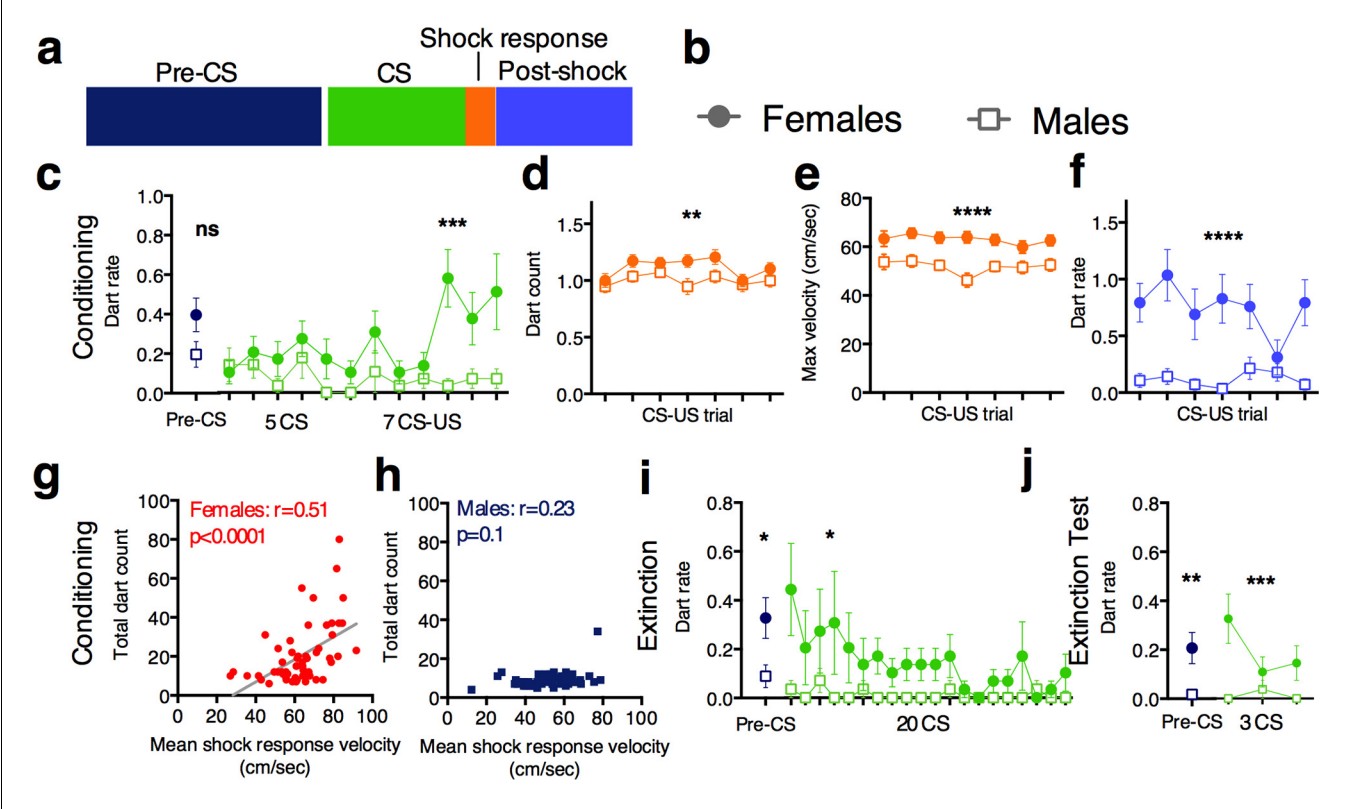

**Figure 2.** Sex differences in darting responses during fear conditioning and extinction. (**a**) The 4 fear conditioning epochs in which velocity was recorded. Graphs in c-f and i-j are color coded to match, and represent mean +/- SEM. (**b**) In graphs c-f and i-j, females are represented by filled circle, males by an open square. (**c**) Pre-CS (final 60 sec before 1st CS presentation) and CS dart rate (darts/min) during conditioning. (**d**) number of darts observed during 5s shock (US) response periods. (**e**) maximum velocity reached during 5s shock (US) response periods. (**f**) mean dart rate observed during 30s post-shock period. (**g**) and (**h**) Pearson's correlations of mean shock response velocity and total session dart count [note that visible male outlier was removed from analysis for being 6 SDs above mean total dart count. When included, r=0.34, p<0.05]. (**i**) Pre-CS and CS dart rate (darts/min) during Extinction. (**j**) Pre-CS and CS dart rate (darts/min) during Extinction testing. *p<0.05; **p<0.01; *** p<0.001; ****p<0.0001 males vs. females.

versus females. Further work will be necessary to determine whether the neurobiological basis of darting is comparable in males and females.

In summary, our data show that during auditory fear conditioning, a substantial subpopulation of predominantly female rats exhibit an active conditioned response associated with reduced conditioned freezing throughout fear conditioning and extinction tests. To our knowledge, this is the first formal characterization of conditioned escape-like responses during classical fear conditioning, in which the shock cannot be avoided. In contrast, learned escape behavior has been well studied in Active Avoidance (AA) paradigms (*Galatzer-Levy et al., 2014*; *Martinez et al., 2013*), and although research into potential sex differences in AA is rare, females are reported to learn AA faster than males (*Dalla and Shors, 2009*), which is consistent with females preferring active fear responses over freezing.

One potentially provocative finding here is that female Darters exhibited comparable freezing to Non-darters at the start of extinction, but enhanced extinction retention the following day. Importantly, lower freezing during extinction retention could not be explained by increased darting during this phase. This suggests that darting during fear conditioning does not interfere with the formation or memory of the tone-shock association, but may confer a long-term protective or adaptive state that promotes increased cognitive flexibility and thus enhanced extinction maintenance (*Maren et al., 2013*). This effect is reminiscent of reports from Maier and colleagues, who have convincingly demonstrated that perceived "escapability" in a shock stress paradigm leads to enhanced AA in subsequent testing (reviewed in *Maier, 2015*). In a similar vein, increases in "active coping"

behavior (digging, rearing, wall-sniffing) during a cued fear memory test are positively correlated

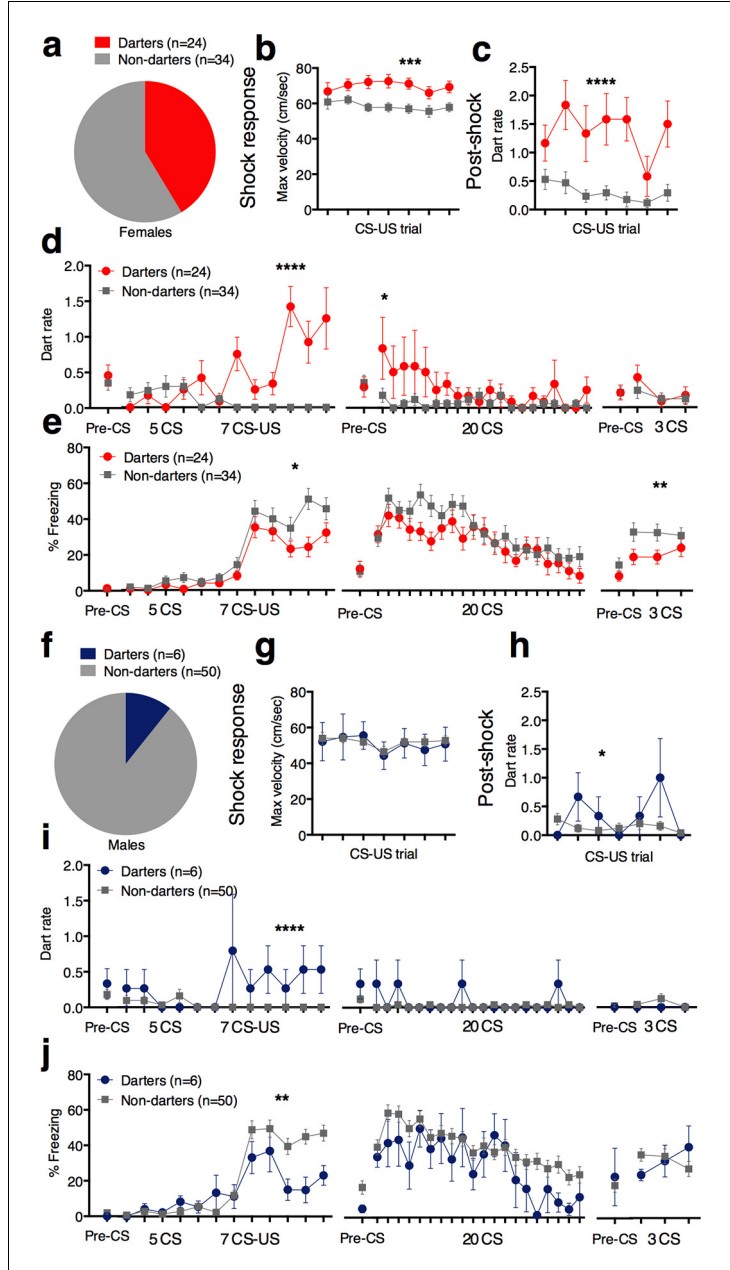

**Figure 3.** Darting subpopulations are greater in females and exhibit distinct behavioral patterns. (a) and (f) proportion of females and males that qualified as Darters. (b) max velocity reached during shock response period (c) mean dart rate (darts/min) observed during 30s post-shock period. (d) Pre-CS and CS dart rate (darts/min) during conditioning, extinction, and extinction test. (e) CS freezing in female Darters vs. Non-darters. (g) Shock response velocity did not differ between male Darters and Non-darters. (h) mean dart rate (darts/min) observed during 30s post-shock period. (i) Pre-CS and CS dart rate (darts/min) during conditioning, extinction, and extinction test. (j) CS freezing in male Darters vs. Non-darters. *$p<0.05$; **$p<0.01$; *** $p<0.001$; ****$p<0.0001$ Darters vs. Non-darters

The following figure supplements are available for Figure 3:

**Figure supplement 1.** Distribution of animals in each estrous cycle phase did not differ between Darters and Non-darters.

with AA success (*Metna-Laurent et al., 2012*). Recruitment of these active coping fear responses instead of freezing has been shown to involve neural transmission in the central amygdala (*Gozzi et al., 2010*) and depend on cannabinoid signaling (*Metna-Laurent et al., 2012*), but to date have not been studied in female rodents. Importantly, these responses have not been demonstrated during fear conditioning learning, the stage at which darting appears to be most critical. Clearly, a great deal of work remains to dissect the neurobiological mechanisms that mediate darting, and to determine its relevance to other indices of active coping, especially in female model organisms.

The finding that conditioned darting occurs primarily in females holds major implications for the interpretation of fear conditioning and extinction studies that use both male and female rats, suggesting that freezing alone may not be a complete measure of learned fear in female subjects. Specifically, female rats that exhibit low freezing levels during fear conditioning could be erroneously described as expressing low fear and/or poor learning, when in fact they have engaged darting responses. This phenomenon may also be clinically relevant, pointing to a sex-specific threat response that predicts enhanced extinction maintenance. Because the learning processes that underlie extinction form the basis for exposure therapy (a common treatment for Post-Traumatic Stress Disorder [PTSD]), a better understanding of the mechanisms that drive darting could lead to improved exposure therapy success. Women are at a twofold risk for PTSD compared to men, and thus identification of the neurobiological factors that determine darting in females may provide insight into sex differences in coping strategies, as well as in stress susceptibility and resilience.

## Materials and methods

### Subjects
Young adult (8–10 weeks) male (n=56) and female (n=58) Sprague Dawley rats were individually housed in the Nightingale Animal Facility at Northeastern University on a 12:12 light:dark cycle with access to food and water *ad libitum*. All procedures were conducted in accordance with the National Institutes of Health Guide for the Care and Use of Laboratory Animals and were approved by the Northeastern University Institutional Animal Care And Use Committee. All experimenters were female.

### Estrous cycle monitoring
Females were vaginally swabbed daily for two weeks to ensure normal estrous cycling. Collected cells were smeared on a microscope slide, stained with Nissl, and examined with a light microscope for cytology.

### Behavioral testing
#### Apparatus and stimuli
Rats underwent habituation, fear conditioning and fear extinction as in (*Gruene et al, 2015*) in one of four identical chambers constructed of aluminum and Plexiglas walls (Rat Test Cage, Coulbourn Instruments, Allentown, PA), with metal stainless steel rod flooring that was attached to a shock generator (Model H13–15; Coulbourn Instruments). The chambers were lit with a single house light, and each chamber was enclosed within a sound-isolation cubicle (Model H10–24A; Coulbourn Instruments). An infrared digital camera allowed videotaping during behavioral procedures. Chamber grid floors, trays, and walls were thoroughly cleaned with water and dried between sessions. Rats were allowed to freely explore the chamber for 4 min before tone presentation on each day began.

#### Fear conditioning
After a 4-minute acclimation period, all rats were exposed to five tone (CS) presentations (habituation), followed by seven conditioning trials (CS–US pairings) on day 1. The CS was a 30-s, 5 kHz, 80 dB SPL sine wave tone, which co-terminated with a 0.5-s, 0.7 mA footshock US during fear conditioning. Mean intertrial interval was 4 min (2–6 min range) throughout habituation and fear conditioning. Freezing was continuously recorded during the conditioning session and analyzed using FreezeFrame Software. Minimum bout was set at 2sec. After conditioning, rats were returned to their home cages.

## Extinction

Freezing was recorded continuously during the extinction training (20 CS presentations, day 2) and test sessions (3 CS presentations, day 3). Both extinction training and testing took place in the same chamber as fear conditioning, but with different contextual cues (floor, light, and odor). Mean inter-trial interval was 4min (2–6 min range).

## Locomotor activity analysis

Video files from FreezeFrame were extracted as QuickTime File Format (.mov) and then converted to MPEG-2 files using AVS Video Converter 9.1 (Online Media Technologies LTD. 2014). The MPEG-2 files were then run through EthoVision software (Noldus), with a center point tracking with a velocity sampling rate of 3.75. Velocity data were computed by Noldus software at 3.75 Hz sampling rate and exported to Matlab (Mathworks). Darts were detected in the exported trace using the *findpeaks* function with a minimum velocity of 23.5 cm/s and a minimum interpeak interval of 0.8 s. The 23.5 cm/s threshold for darts was determined by cross-referencing velocity data with experimenter scoring of darting behavior. 23.5 cm/s was the velocity at which all movements at that rate or higher were consistently scored as darts. These discrete events were registered to each trial and analyzed with custom Matlab software (available as a *source code 1* file).

## Statistical analysis

Darting, velocity, and freezing values during each epoch were averaged for each group and analyzed for each session (fear conditioning, extinction, extinction test) using 2-way repeated measure ANOVAs with factors of group and trial. Mann-Whitney t-tests were used for all Pre-CS comparisons. One male animal was removed from analysis because its total dart count was 6 standard deviations outside the mean (shown in *Figure 2h*).

## Acknowledgments

We thank Michael Gunson, Jesse Katon, Marc Lowe, and Heather Brenhouse for technical support, and Mark Baxter for manuscript comments.

## Additional information

### Funding

| Funder | Grant reference number | Author |
|---|---|---|
| National Institute of Mental Health | R21 MH098006-01 | Rebecca M Shansky |

The funders had no role in study design, data collection and interpretation, or the decision to submit the work for publication.

### Author contributions

TMG, Conception and design, Acquisition of data, Analysis and interpretation of data, Drafting or revising the article; KF, Acquisition of data, Analysis and interpretation of data, Drafting or revising the article; AS, Acquisition of data, Contributed unpublished essential data or reagents, Drafting or revising the article; SDS, Analysis and interpretation of data, Drafting or revising the article; RMS, Conception and design, Analysis and interpretation of data, Drafting or revising the article

### Author ORCIDs

Stephen D Shea, http://orcid.org/0000-0001-6927-0189

### Ethics

Animal experimentation: All procedures were conducted in accordance with the National Institutes of Health Guide for the Care and Use of Laboratory Animals and were approved by the Northeastern University Institutional Animal Care and Use Committee protocol # 12-0102R.

## Additional files

**Supplementary files**
• Source code 1. The Matlab script used to detect and analyze darts is available here.

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
