## [Decision Letter]

Thank you for submitting your work entitled "Sexually divergent expression of active and passive conditioned fear responses in rats" for consideration by *eLife*. Your article has been favorably evaluated by a Senior Editor and four peer reviewers, one of whom, Peggy Mason, is a member of our Board of Reviewing Editors. One the four reviewers has agreed to reveal her identity: Margaret McCarthy.

The reviewers have discussed the reviews with one another and the Reviewing editor has drafted this decision to help you prepare a revised submission.

Summary:

All reviewers believe this is an important and timely topic that has the potential to greatly advance our understanding of hitherto contradictory results and ultimately help to accurately illuminate the effects of fear on behavior. It was broadly felt that the information provided will be of substantial use to researchers in the U.S. who are required (by NIH) to incorporate sex as a biological variable in all preclinical research.

Essential revisions:

1) Better define darting and darters. What constitutes darting? How much darting at what time points makes a rat into a darter? Justify the criteria chosen for defining darters. Are the male darters really darters or just fast-moving dart-impostors that do not show the full phenotype that female darters show? The graphs suggest that the male darters are quite different from the female ones.

2) The design of the study is a bit unusual. There are 5 CS alone presentations prior to conditioning aimed at measuring habituation to the CS. However, this design induces latent inhibition slowing down the acquisition of the CS-US association. However, the interpretation by the authors that a lack of darting during the CS-only trials indicates that darting responses reflect conditioning is a bit problematic. The authors should also explain the unusual design with 5 CS presentations as well as the choice to not use unpaired CS-US presentations as controls (the typical approach).

3) A longer Discussion is needed to pick up the points raised throughout the Results but never discussed (e.g. the suggestion that darting is an adaptive response linked to positive outcomes and not simply competition for shared motor resources).

4) Work by Gozzi et al. (2010) and Laurent and colleagues (2012) demonstrate that conditioned freezing behaviors are inhibited to favor active coping strategies in classical fear conditioning paradigms. Thus, the temporal decrease in freezing behaviors observed in both male and female rats during extinction should be accompanied by an increase in active coping behaviors in these experimental animals. Did the authors not observe any switch from passive to active behaviors, and vice versa, in these experimental animals? Were other active coping strategies not present in male mice that were distinct from the darting phenotype present in female rats? Did female mice not exhibit other forms of active coping strategies other than darting? Alternatively, if the authors are arguing that their observed "darting" is different form of fear behavior than that of active or passive, it would serve the authors well to reconcile these claims with the current literature. In sum, please place the present results into the context of active vs passive coping strategies, adaptive strategies, and why those may differ between males and females.

5) Examine whether there are individual differences in darting that may be a function of weight or any available measure of locomotion such as open field. A number of additional specific suggestions would strengthen the manuscript. As one example, a specific comparison of freezing and darting across trials would be of interest (see below).

6) Could the authors include a movie of darting?

Minor Comments:

Reviewer #1:

Given the reports of the effect of male hormones (from experimenters) on rodent behavior (e.g. Mogil work), please specify the gender of the experimenters.

Please explicitly state that Figure 1 and Figure 2 include all males/females (not just the darters), if that is the case. In any case clarify.

Please clarify the following passage: "Compared to non-darters, female darters exhibited greater shock response velocities […] as well as higher dart rates in the post-shock period." Darters are defined by their darting rates during the post-shock period. If the reviewer is correct, remove this. If the reviewer is wrong, please explain why.

Reviewer #2:

1) One can't help notice that the darting behavior by females learning to fear the stimulus sounds remarkably similar to the "hopping and darting" that sexually receptive females show to solicit male attention. Can the authors comment on this and do they have any way of comparing these two darting behaviors to see if they are in fact similar or distinct?

2) Wouldn't you expect dart velocity to be related to body weight? And is that why females have a higher velocity?

Reviewer #3:

It would be useful to compare the darting behavior to freezing behavior more directly in Figure 1. Specifically, it would be helpful to see similar graphics for Figure 1, and e for freezing behavior to visualize the similarity and differences in darting and freezing across training trials.

It would be interesting to know if previous studies have found that males and females have more similar acquisition rates for conditioned suppression (where a decrease in bar pressing for a food reward is a measure of fear) than for fear conditioning. If so, that would further support the argument that there are better measures of fear in females.

Reviewer #4:

In Figure 1, female rats exhibit greater darts/min as compared to male rats in control conditions (CS only) and prior to fear conditioning. Though the authors argue that there are no within-sex differences in dart rates between darters and non-darters prior to conditioning, it would serve the authors well to examine whether there are locomotor differences between female and male cohorts prior to fear conditioning. This will help the authors to determine whether pre-existing activity differences predispose rats to exhibiting a darting phenotype as a learned fear behavior.

To strengthen their argument, the authors should not only compare dart rate extinction between female and male rats but they should also compare the freezing behaviors for both male and female rats as well (Figure 2).

In Figure 3, there are no significant differences in the dart rate during extinction training (20CS) yet significance is indicated (****).

For the supplemental data, please indicate the statistic test that was carried out to "determine there was no effect of the estrous cycle on darting”.

---

## [Author Response]

*Essential revisions:*

*1) Better define darting and darters. What constitutes darting? How much darting at what time points makes a rat into a darter? Justify the criteria chosen for defining darters.*

In our response to this set of questions, we feel it is important to distinguish between our definition of a dart (as a discrete event), a darter in the narrow sense (an individual that was included in our darting analysis group) and a darter in the broad sense (a ‘true darter’ as opposed to an ‘imposter’). We have crisply defined the first two entities, however it was not our goal to delineate true darting. We completely agree with the reviewer that this is a very interesting question, but we simply note in the Discussion that there are crucial features to the darting exhibited by females that are not seen in males. The implication is that male darting, as observed in this study, is qualitatively different from female darting, however we did not attempt to define or segregate true darters.

We have now expanded the text of the manuscript to more clearly define the criteria we used to define both darting and Darters (Results and Discussion, as well as in “Locomotor activity analysis” section of Materials and methods). In addition, we include a video (Video 1) of an example of darting. In order to be classified as a Darter, an animal must have exhibited at least one dart during fear conditioning CS (tone presentation) 8-12. Animals that darted in response to shock or during inter-trial intervals but not during the CS were not considered Darters. Because darting during fear conditioning has not been previously characterized, we chose to set a relatively inclusive criterion. As we move forward with our work and begin to dissect the neurobiological basis of darting, we expect that the criteria may be refined.

*Are the male darters really darters or just fast-moving dart-impostors that do not show the full phenotype that female darters show? The graphs suggest that the male darters are quite different from the female ones.*

Although the male Darters reach the criterion of one dart during late fear conditioning CS presentations, we agree that they do not exhibit many of the other characteristics we observe in female Darters (as discussed above). These behavioral differences between male and female Darters are discussed in paragraph six of Results and Discussion, but clearly further work is needed to determine the functional significance, if any, of darting in males.

*2) The design of the study is a bit unusual. There are 5 CS alone presentations prior to conditioning aimed at measuring habituation to the CS. However, this design induces latent inhibition slowing down the acquisition of the CS-US association.*

This is an interesting and important point. Although latent inhibition can be induced by prior exposure to a CS, we do not feel that it is a significant risk with our particular preparation. To date, latent inhibition in fear conditioned rats has only been demonstrated when CS pre-exposure is considerable and CS-US pairings brief, e.g. a pre-exposure of 20 or 30 tone presentations prior to a single CS-US presentation (De la Casa, 2013; Rudy, 1994), when tone pre-exposure is protracted in duration (Baker et al., 2012), or when it occurs over several sessions (Sotty et al., 1996). In other words, it appears that in order for latent inhibition to occur in cued fear conditioning, CS pre-exposure must be far more robust than 5 30-sec CSs, especially given the fairly high number of CS-US pairings (7) in our preparation. Importantly, we have used this design in three recent publications (Gruene et al., 2014, 2015; Rey et al., 2014) and reliably observe rapid increases in freezing to CS during fear conditioning learning and testing, suggesting that latent inhibition plays a minimal, if any, role in our behavioral outcomes.

*However, the interpretation by the authors that a lack of darting during the CS-only trials indicates that darting responses reflect conditioning is a bit problematic.*

We apologize if our interpretation of the lack of CS-only darting was unclear or overstated. While those data alone cannot indicate whether darting is a conditioned response, we infer that at the very least, darting is not an unconditioned response to the tone, i.e. there is nothing inherent about the tone that elicits darting in Darters, nor are Darters simply “jumpy” animals. We have added to the manuscript text (paragraph three, Results and Discussion) to clarify this point. However, we feel that the most convincing evidence that darting is a conditioned response is the fact that CS-specific darting can be observed in female Darters at the start of extinction (Figure 3), when the animal does not experience any footshocks. This response is predicted by the presence of darting during conditioning. It is also important to note that during conditioning, female Darters show increased propensity to dart specifically during the tone. We argue that these observations collectively constitute strong evidence that darting is a conditioned response.

*The authors should also explain the unusual design with 5 CS presentations as well as the choice to not use unpaired CS-US presentations as controls (the typical approach).*

This design was based on numerous publications from experts in the field (Burgos-Robles et al., 2007; Bush et al., 2007; Galatzer-Levy et al., 2013; Milad and Quirk, 2002; Sotres-Bayon et al., 2007), and is fairly common for studies in which extinction of cued fear is a focus.

We want to emphasize that the 5 CS habituation tones were not intended to represent a “control” condition, but merely a way of establishing baseline responsiveness to the CS prior to its association with the US. We agree that a thorough examination of darting in alternate CS-US arrangements will be an important next step in defining the situational boundaries of darting. In an unpaired CS-US design, the CS would signal safety (Pollak et al., 2010), and thus we would not expect to observe increased darting to the CS as we do in a paired design here. But because of the large scale of the current study, the novelty of the sex-specific findings within, and the topical nature of these findings with respect to recent changes in NIH policy, we felt it was appropriate to report the current findings before undertaking another large-scale study.

*3) A longer Discussion is needed to pick up the points raised throughout the Results but never discussed (e.g. the suggestion that darting is an adaptive response linked to positive outcomes and not simply competition for shared motor resources).*

We have now expanded the Discussion to elaborate on these points (please see paragraph eight).

*4) Work by Gozzi et al. (2010) and Laurent and colleagues (2012) demonstrate that conditioned freezing behaviors are inhibited to favor active coping strategies in classical fear conditioning paradigms. Thus, the temporal decrease in freezing behaviors observed in both male and female rats during extinction should be accompanied by an increase in active coping behaviors in these experimental animals. Did the authors not observe any switch from passive to active behaviors, and vice versa, in these experimental animals?*

We have read both of these excellent papers carefully, and while on the surface they appear to be relevant to the current study because they also deal with active and passive behaviors, there are considerable distinctions in design and behavioral measures that make direct extrapolation to our work difficult. During extinction, an animal learns not to fear the CS. We would therefore expect all forms of fear responses (both passive and active) to subside. This is indeed what we observe, as can be seen in Figure 3. It’s worth noting that in both referenced papers, animals underwent a comparably weak conditioning procedure – either “partial” conditioning (Gozzi et al., 2010) or a single CS-US pairing (Metna Laurent et al., 2012) vs. our 7 CS-US pairings – and were tested for fear memory/extinction during a single 6- or 8-minute CS exposure, vs. our 90 minute, 20 CS extinction session. It is likely that the differences in CS and US parameters influenced the behavioral strategies and trajectories animals engaged in those studies and ours. That said, we would predict that had the extinction sessions in both papers been substantially longer, they would have also observed a decrease in active responding.

*Were other active coping strategies not present in male mice that were distinct from the darting phenotype present in female rats? Did female mice not exhibit other forms of active coping strategies other than darting? Alternatively, if the authors are arguing that their observed "darting" is different form of fear behavior than that of active or passive, it would serve the authors well to reconcile these claims with the current literature.*

Active coping behaviors in both papers were similarly defined as “digging, exploration, and rearing,” or “digging, rearing, and wall-sniffing,” none of which would have qualified as darting. Because the specific goal of the current manuscript is to define darting as a conditioned fear response, we did not evaluate these other active coping behaviors (some, such as digging, were not possible because there was no bedding in the test chambers). However, a thorough characterization of all measurable behaviors during fear conditioning and extinction in the future will undoubtedly be informative.

It is also important to note that the test phase in which behavioral analysis was focused is different between these papers and the current study. In both Gozzi et al and Metna-Laurent et al, behaviors were measured only during the fear memory/extinction phase (i.e. Day 2), and behavior during fear conditioning itself is not reported. In contrast, the key darting phase in our analysis is fear conditioning (Day 1) – Darters are defined based on darting behavior during this phase, and our hypothesis is that the engagement of darting during fear conditioning elicits long term changes that influence behavior on subsequent test days. We look forward to testing this hypothesis in follow-up studies.

Finally, subjects in the referenced studies were exclusively male mice, so it is not known whether their observations would have carried to female mice, let alone female rats, as were used in the current study.

*In sum, please place the present results into the context of active vs passive coping strategies, adaptive strategies, and why those may differ between males and females.*

We have added to the Discussion of the manuscript to address the relevance of our findings to these papers and others (paragraph eight).

*5) Examine whether there are individual differences in darting that may be a function of weight or any available measure of locomotion such as open field.*

We have provided data to address these issues below, in response to Minor Points from Reviewer 2 (weight) and Reviewer 4 (locomotion).

*6) Could the authors include a movie of darting?*

We now include a movie of darting (Video 1).

Minor Comments:

Reviewer #1:

*Given the reports of the effect of male hormones (from experimenters) on rodent behavior (e.g. Mogil work), please specify the gender of the experimenters.*

We now provide this information in the Materials and methods, under “subjects.” All experimenters were female.

*Please explicitly state that Figure 1 and Figure 2 include all males/females (not just the darters), if that is the case. In any case clarify.*

We now explicitly state this in the manuscript text, in addition to the Figure caption. All 114 animals (58 females, 56 males) are represented in all figures. It is only in Figure 3 that animals are separated into “Darters” and “Non-darters.”

*Please clarify the following passage: "Compared to non-darters, female darters exhibited greater shock response velocities […] as well as higher dart rates in the post-shock period." Darters are defined by their darting rates during the post-shock period. If the reviewer is correct, remove this. If the reviewer is wrong, please explain why.*

We apologize that the criteria for Darting were unclear. Darters were categorized exclusively based on whether they darted during the CS (tone presentation), not on their dart rate during the postshock period, in which there is no CS. We then evaluated shock response velocities and postshock dart rate in these groups (Figure 3). As noted above in our response to comment 1, we have expanded our explanation of both darting and Darter criteria and hope this clarifies the issue.

Reviewer #2:

1) One can't help notice that the darting behavior by females learning to fear the stimulus sounds remarkably similar to the "hopping and darting" that sexually receptive females show to solicit male attention. Can the authors comment on this and do they have any way of comparing these two darting behaviors to see if they are in fact similar or distinct?

This is an interesting question. On a strictly locomotor basis, the darts we observe here are fairly similar in appearance to those during solicitation. It is of course not yet known whether the neural mechanisms that elicit darting in these two vastly different scenarios overlap. However, we do not observe any “hopping” behaviors that resemble those of solicitation behavior (Erskine, 1989). Moreover, we observe conditioned darting in females of all estrous stages (see Figure 3—figure supplement 1) as well as in a few males, suggesting that conditioned darting is not simply indicative of female proceptivity. There is also no aspect of the task that could be expected to elicit sexual behaviors in females (e.g. there is no male or male pheromones present during the behavior).

2)Wouldn't you expect dart velocity to be related to body weight? And is that why females have a higher velocity?

It makes sense that the lighter an animal is, the more easily it could move around, thus making it more likely to dart. To address the possibility that lower weight predisposed animals to darting, we compared body weight between Darters and Non-Darters, but found no significant differences (Figure 4). Although males clearly weigh more than females, we feel these data suggest that weight alone cannot account for the incidence of darting.

**Author response image 1. fig4:** **DOI:**
http://dx.doi.org/10.7554/eLife.11352.009

Reviewer #3:

It would be useful to compare the darting behavior to freezing behavior more directly in Figure 1. Specifically, it would be helpful to see similar graphics for Figure 1, and e for freezing behavior to visualize the similarity and differences in darting and freezing across training trials.

We agree that comparing darting and freezing side-by-side would be informative. However, the visual representations of darting in Figure 1 depict event frequencies for male and female populations at given time points. Because freezing is not a discrete event, it does not lend itself to this form of representations. We feel that the best comparison of freezing and darting can be observed in Figure 3, which intentionally shows darting and freezing directly above one another. In this way, you can see that Darters’ and Non-darters’ trajectories of darting and freezing are inverse.

*It would be interesting to know if previous studies have found that males and females have more similar acquisition rates for conditioned suppression (where a decrease in bar pressing for a food reward is a measure of fear) than for fear conditioning. If so, that would further support the argument that there are better measures of fear in females.*

To date, there is very little existing data on sex differences in conditioned suppression. However, Maes (2002) did not find a sex difference in the magnitude of conditioned fear suppression. It is worth noting that in Active Avoidance studies, females have shown enhanced learning compared to males (Dalla and Shors, 2009). We now include these studies in the Discussion.

Reviewer #4:

In Figure 1, female rats exhibit greater darts/min as compared to male rats in control conditions (CS only) and prior to fear conditioning. Though the authors argue that there are no within-sex differences in dart rates between darters and non-darters prior to conditioning, it would serve the authors well to examine whether there are locomotor differences between female and male cohorts prior to fear conditioning. This will help the authors to determine whether pre-existing activity differences predispose rats to exhibiting a darting phenotype as a learned fear behavior.

As we note in the second paragraph of the manuscript Introduction, there is a vast literature demonstrating heightened locomotor activity in females compared to males. When we examine baseline pre-CS locomotor activity (measured as mean velocity in cm/s), we also observe higher mean velocities in females ( Figure 5, p<0.0001, unpaired t-test). However, baseline locomotor activity was not different between Darters and Non-darters in either sex, suggesting that locomotor activity does not predict the likelihood of darting.

**Author response image 2. fig5:** **DOI:**
http://dx.doi.org/10.7554/eLife.11352.010

To strengthen their argument, the authors should not only compare dart rate extinction between female and male rats but they should also compare the freezing behaviors for both male and female rats as well (Figure 2).

We have previously compared freezing during extinction in males and females, and find no overall sex differences in extinction (Rey et Al., 2014; Gruene et al., 2015). Although general sex differences in freezing during extinction was not the focus of the current manuscript, we have provided those data below. We find a slight but significant main effect of sex in freezing extinction learning (F_1,112_=4.02, p=0.048, 2-way repeated measures ANOVA). We believe the discrepancy between this result and our previous findings is due to the considerable statistical power conferred by n’s of ~60/sex. We do not observe an effect of sex in extinction retrieval (F_1,112_=1.84, p=0.18, 2-way repeated measures ANOVA). Data are represented in Figure 6.

**Author response image 3. fig6:** **DOI:**
http://dx.doi.org/10.7554/eLife.11352.011

In Figure 3, there are no significant differences in the dart rate during extinction training (20CS) yet significance is indicated (****).

We have fixed this error.

*For the supplemental data, please indicate the statistic test that was carried out to "determine there was no effect of the estrous cycle on darting”.*

We ran a chi-square test to identify differences in the estrous cycle distribution between Darters and Non-darters, which returned a value of 2.785, p=0.42. This information has been added to the manuscript.